# The Pharmacological Inhibition of CaMKII Regulates Sodium Chloride Cotransporter Activity in mDCT15 Cells

**DOI:** 10.3390/biology10121335

**Published:** 2021-12-16

**Authors:** Mohammed F. Gholam, Benjamin Ko, Zinah M. Ghazi, Robert S. Hoover, Abdel A. Alli

**Affiliations:** 1Department of Physiology and Functional Genomics, University of Florida College of Medicine, Gainesville, FL 32610, USA; mohammed.gholam@ufl.edu; 2Department of Basic Medical Sciences, King Saud Bin Abdulaziz University for Health Sciences, Jeddah 21423, Saudi Arabia; 3Department of Medicine, University of Chicago, Chicago, IL 60637, USA; bko@medicine.bsd.uchicago.edu; 4Department of Physiology, Emory University School of Medicine, Atlanta, GA 30322, USA; zghazi@emory.edu; 5Section of Nephrology and Hypertension, Department of Medicine, Tulane University School of Medicine, New Orleans, LA 70112, USA; rhoover3@tulane.edu

**Keywords:** NCC, filamin A, CaMKII

## Abstract

**Simple Summary:**

The renal sodium chloride cotransporter (NCC) plays an important role in the total body electrolyte balance and blood pressure control. The regulation of this protein by the actin cytoskeleton has not been thoroughly studied. Here, we investigate a novel association between the actin cytoskeleton protein filamin A and the NCC using a mouse cellular model and in the native kidney. Our results show for the first time that the disruption of the actin cytoskeleton reduces NCC activity and filamin A plays an essential role in NCC protein expression in cells of the distal convoluted tubule. We further show that the pharmacological inhibition of the Ca^2+^/calmodulin dependent protein kinase II (CAMKII) augments NCC protein expression. These results introduce a new mechanism for the regulation of the NCC.

**Abstract:**

The thiazide-sensitive sodium chloride cotransporter (NCC) in the distal convoluted tubule is responsible for reabsorbing up to one-tenth of the total filtered load of sodium in the kidney. The actin cytoskeleton is thought to regulate various transport proteins in the kidney but the regulation of the NCC by the actin cytoskeleton is largely unknown. Here, we identify a direct interaction between the NCC and the cytoskeletal protein filamin A in mouse distal convoluted tubule (mDCT15) cells and in the native kidney. We show that the disruption of the actin cytoskeleton by two different mechanisms downregulates NCC activity. As filamin A is a substrate of the Ca^2+^/calmodulin-dependent protein kinase II (CaMKII), we investigate the physiological significance of CaMKII inhibition on NCC luminal membrane protein expression and NCC activity in mDCT15 cells. The pharmacological inhibition of CaMKII with the compound KN93 increases the active form of the NCC (phospho-NCC) at the luminal membrane and also increases NCC activity in mDCT15 cells. These data suggest that the interaction between the NCC and filamin A is dependent on CaMKII activity, which may serve as a feedback mechanism to maintain basal levels of NCC activity in the distal nephron.

## 1. Introduction

Hypertension is a growing public health concern worldwide and is a major risk factor for increasing morbidity and mortality from strokes, cardiovascular diseases, and renal failure [1]. However, it often remains clinically silent until complications arise. Hypertension affects more than 40% of adults older than the age of 25 years and nearly one-third of the adult population in the United States [2]. Moreover, it has been estimated that it will affect 1.56 billion individuals by the year 2025 [3]. In addition to its high prevalence, the molecular pathogenesis of the most common forms of hypertension is still not fully understood.

Salt balance regulation by the kidney is the cornerstone for blood pressure regulation by adjusting the blood volume in response to changes in systemic pressure [4]. The kidneys play an important role in blood pressure regulation through renal transporters and ion channels along the nephron. The distal convoluted tubule (DCT) is part of the aldosterone-sensitive distal nephron and it plays a critical role in maintaining the total body sodium balance and blood pressure control [5]. Although it is the shortest segment in the nephron, it plays a crucial role in a variety of homeostatic processes including sodium chloride reabsorption, potassium secretion, and calcium handling. The DCT reabsorbs 5 to 10% of the filtered sodium through the electroneutral thiazide-sensitive sodium chloride cotransporter (NCC) [5,6]. The abnormal expression and activity of the NCC have been implicated in blood pressure and electrolyte imbalances including Gitelman and Gordon syndromes [5,6]. The NCC is regulated by the with-no-lysine kinase (WNK)-oxidative stress-responsive 1 (OSR1)/Ste-like proline/alanine-rich kinase (SPAK)-NCC phosphorylation cascade [6]. The activation of this pathway results in an increased phosphorylation of the NCC and, thus, a higher NCC activity [6].

The mammalian cytoskeleton is composed of three major protein families including microfilaments, microtubules, and intermediate filaments as well as several cytoskeleton-associated proteins [7]. The actin cytoskeleton and its associated proteins plays a crucial role in the cell shape, motility, signal transduction, vesicular trafficking, and function of ion channels and transporters [7]. Moreover, it is known to regulate numerous epithelial transport proteins including the renal epithelial sodium channels (ENaC) [8,9,10,11] and aquaporin 2 (AQP2) [12] as well as chloride channel-2 (CIC-2) and -3 (CIC-3) [13,14]. Our lab was among the first to show that two actin cytoskeleton proteins, specifically MARCKS and filamin, positively regulate the renal epithelial sodium channel (ENaC) [9]. Furthermore, one report by Dimke and colleagues showed that the actin cytoskeletal protein γ-adducin directly binds to the amino terminus of the renal NCC in an unphosphorylated state and stimulates its activity [15]. However, it is not known if the NCC associates with other actin cytoskeleton proteins and the mechanism of regulation of the NCC is still not fully understood.

Here, we investigated the interaction between the actin cytoskeleton protein filamin A and the renal NCC and whether filamin A regulates the density of the NCC at the luminal membrane. Filamins are large actin binding phosphoproteins that scaffold the microfilaments immediately beneath the cell membrane [16]. They also play a critical role in stabilizing the cell membrane, particularly during changes in cell shapes associated with cell motility and migration [16]. It was found that multiple integral membrane proteins including CFTR, ß-integrins, dopamine receptors, and Kv4.2 potassium channels interact with filamin proteins [17,18]. However, the association between filamin A and the NCC in the DCT has not been reported. Therefore, we hypothesized that filamin A positively regulated the NCC in the distal convoluted tubule by maintaining NCC phosphorylation at the luminal membrane. We investigated this novel mechanism in cultured mouse distal convoluted tubule cells. Taken together, this study introduces a novel association between the NCC and the actin cytoskeleton and addresses the physiological significance of this interaction.

## 2. Materials and Methods

### 2.1. Cell Culture

Mouse DCT15 cells (mDCT15), previously characterized by Ko et al. [19], were cultured in a 1:1 mixture of DMEM-F-12 media (Corning; Manassas, VA, USA) supplemented with 5% heat-inactivated fetal bovine serum and 1% penicillin-streptomycin-neomycin at 37 °C. The cells were used for experiments when they reached a 90% confluence.

### 2.2. SiRNA Transduction

Negative control non-targeting siRNA (siGENOME Non-Targeting siRNA Pool: D-001206-14-05) and filamin A siRNA (siGENOME SMART Pool: #M-058520-02-0005) were prepared according to the manufacturer’s instructions (Dharmacon, Inc.; Chicago, IL, USA). The transfection of siRNA was performed using DharmaFECT 1 Transfection Reagent (Dharmacon, Inc.; Chicago, IL, USA) when the cells reached a 70% confluency.

### 2.3. Immunofluorescence

The mDCT15 cells were fixed in a 1:1 solution of methanol:acetone (*v*/*v*) for exactly 10 min in a freezer at −20 °C. A series of three washes with a Hanks balanced salt solution (HBBS) (Thermo Fisher Scientific; Gibco; Waltham, MA, USA) was performed followed by blocking with normal goat serum prepared in 1XPBS for 20 min at room temperature. The blocking solution was replaced with a 1:1000 dilution of a pNCC (phospho Solution# p1311-53) primary antibody prepared in normal goat serum and 1XPBS before the cells were incubated for 45 min at room temperature. After a series of three HBBS washes, the cells were incubated in a solution of a 488 donkey anti-rabbit secondary antibody (Invitrogen# 1874771) (1:2000) prepared in 1XPBS and normal goat serum and then incubated for 30 min at room temperature. Finally, the cells were subject to a series of three washes with HBSS before being imaged by fluorescent microscopy (Eclipse Ti2 by Nikon Instruments Inc., Melville, NY, USA).

### 2.4. Mice

The animal studies were approved by the Emory University and the University of Florida Institutional Animal Care and Use Committees and were in compliance with the National Institutes of Health “Guide for the Care and Use of Laboratory Animals”. Adult C57B6 wild-type mice were euthanized and the kidneys were homogenized and used for immunoprecipitation and Western blot studies.

### 2.5. Immunoprecipitation

The cells were rinsed once with 1XPBS before being lysed in an ice-cold mammalian protein extraction reagent (MPER) containing Halt protease and phosphatase inhibitors (Thermo Fisher Scientific). The cell lysates were incubated on ice for 1 h with gentle mixing every 10 min. For each immunoprecipitation reaction, 300 μL of cell lysate was added to a clean microcentrifuge tube before adding 5 μL of a primary NCC or filamin antibody (Cell Signaling Tech; Danvers, MA, USA). The tubes were incubated overnight with end-over-end rocking at 4 °C. A total of 50 μL of a 50% protein G bead slurry (Thermo Fisher Sci) prewashed with 1XPBS was then added and the tubes were subject to end-over-end rocking for an additional 4 h at 4 °C. The complexes were washed three times with 1XPBS before the bound proteins were eluted by adding Laemmli sample buffer. The samples were then boiled for 10 min. The supernatant was loaded onto 4–20% SDS-PAGE gels for further analysis by Coomassie blue staining or Western blotting.

### 2.6. SDS-PAGE and Western Blotting

The mDCT15 cells were lysed in an ice-cold MPER supplemented with Halt protease and phosphatase inhibitors (Thermo Fisher Scientific, Waltham, MA, USA). The cell lysates were sonicated on ice three times for 5 s intervals. Fifty milligrams of kidney cortex tissue from C57B6 wild-type mice were homogenized using an Omni TH homogenizer (Kennesaw, GA, USA) in a tissue protein extraction reagent (TPER; Thermo Scientific, Waltham, MA, USA) supplemented with protease and phosphatase inhibitors (Thermo Scientific, Waltham, MA, USA). The protein concentration was determined by a bicinchoninic acid assay (Thermo Scientific Pierce, Waltham, MA, USA). One hundred micrograms of the total protein were loaded onto 4–20% Criterion precast gels (BIO-RAD; Hercules, CA, USA) and subject to sodium dodecyl sulphate-polyacrylamide gel electrophoresis. The resolved proteins from the gels were transferred to nitrocellulose membranes. The membranes were blocked in a blocking solution (5% non-fat dry milk in 1XTBS) before being incubated with a 1:1000 dilution of either an anti-NCC (StressMarq SPC-402D), anti-filamin A (Cell Signaling Tech; Danvers, MA, USA), anti-spectrin (Thermo Scientific; Waltham, MA, USA), or anti-phospho-NCC antibody (Ko, 2012 #38) prepared in 5% BSA and 1XTBS overnight at 4 °C. For the generation of the NCC antibody, the amino-terminal phospho-peptide sequence (CGS TLY MR(p)T FGY NT) was provided to Pocono Rabbit Farms and Laboratory for the antigen generation. The peptide was generated, conjugated to keyhole limpet hemocyanin and injected into rabbits according to their protocol. The sequential bleeds were screened by ELISA, yielding an appropriate serum sample. The serum was then affinity purified utilizing a column with the immunizing phospho-peptide. In order to remove the non-phospho-NCC antibody, the eluent from the first purification was then poured on a column bound with a non-phospho-NCC peptide (CGS TLY MRT FGY NT). The resulting flowthrough contained the phospho-specific NCC antibody and immunoblotting was performed to confirm the specificity of the immunopurified antibody. After the primary antibody incubation, the blots were washed with 1XTBS and incubated in a 1:3000 dilution of goat anti-rabbit peroxidase conjugated secondary antibody (BIO-RAD; Hercules, CA, USA) prepared in a blocking solution for 1 h at room temperature. The blots were washed before being incubated for 7 min in SuperSignal West Dura chemiluminescent solution (Thermo Scientific Pierce, Waltham, MA, USA). The blots were developed using a Kodak or BioRad imaging system.

### 2.7. Assessment of the NCC Function in the Cells

Using 12-well plates, the mDCT15 cells were seeded and incubated in serum-free growth media (Opti-Mem) for exactly 24 h before starting the assay. Subsequently, these seeded mDCT15 cells were subjected to a treatment with either KN93 or a vehicle (DMSO) for the indicated times and concentrations. Thirty minutes before the uptake, 0.1 mM metolazone (an inhibitor of the NCC) or the vehicle (DMSO) (Sigma) was then added to the media in order to ensure that the NCC was inhibited during the uptake period. The media was then replaced with a ^22^Na^+^-containing medium (140 mM NaCl or as indicated), 1 mM CaCl, 1 mM MgCl, 5 mM HEPES/Tris pH 7.4, 1 µM amiloride, 0.1 mM bumetanide, 1 µM benzamil, 1 mM ouabain, and 1 microCi/mL of ^22^Na^+^) with or without thiazide (0.1 mM metolazone) and then incubated for exactly 20 min. Thereafter, the tracer was stopped via washes with an ice-cold washing buffer. Subsequently, the cells were lysed in 0.1% SDS (sodium dodecyl sulfate). The radioactivity of the previously lysed cells was measured via liquid scintillation and their protein concentrations were determined (Bicinchoninic Acid (BCA) Protein Assay, Pierce). Eventually, the uptakes were normalized to nmol/mg/min and the thiazide-sensitive uptake was given by the difference of the uptakes with and without thiazide.

### 2.8. Statistical Analysis

A statistical analysis was performed using SigmaPlot software (Systat, San Jose, CA, USA). A Student’s *t*-test was used to compare two groups and a one-way ANOVA was used to compare three or more groups. Data were reported as means ± SE and a *p*-value of less than 0.05 was considered statistically significant.

## 3. Results

### 3.1. Disruption of the Actin Cytoskeleton Decreases NCC Activity in mDCT15 Cells

In order to determine whether the activity of the renal NCC was affected by disruptions in the actin cytoskeleton, we treated mDCT15 cells with two different chemical disruptors of the actin cytoskeleton, latrunculin B and cytochalasin D. These two agents cause a disruption in the actin cytoskeleton by different mechanisms. We refrained from using a high concentration of either agent for prolonged treatment times in order to avoid potential cytotoxic effects. Instead, we used a concentration of 500 nM for latrunculin B and a concentration of 1 μM for cytochalasin D for a duration of 30 min. Both the latrunculin B and cytochalasin D treatment of cultured mDCT15 cells resulted in a decrease in NCC activity compared with the vehicle treatment, as shown in Figure 1. The decrease in NCC activity was greater after the cytochalasin D treatment compared with the latrunculin B treatment despite latrunculin B being 10- to 100-fold more potent than cytochalasin.

### 3.2. siRNA-Mediated Knockdown of Filamin A Reduces the Total NCC and Phospho-NCC Expression in mDCT15 Cells

In order to further confirm the NCC expression and its association with filamin A, we probed for NCC protein expression in mDCT15 cells transfected with either filamin A siRNA or a control siRNA. Transfection efficiency was assessed by Western blotting and immunocytochemistry, as shown in Figure 2. Western blot and densitometric analysis showed a decrease in the total NCC protein expression in the membrane fractions of the mDCT15 cells compared with the cells transfected with the control siRNA (Figure 3). We then investigated whether the protein expression of the active form of NCC, phospho-NCC, was regulated by filamin A. We performed similar studies but probed for phospho-NCC by a Western blot. The Western blot and densitometric analysis, as shown in Figure 4, showed a decrease in phospho-NCC, similar to the total NCC after the siRNA-mediated knockdown of filamin A. In order to corroborate the Western blot results for phospho-NCC, we performed immunofluorescence studies. The mDCT15 cells were transfected with filamin A siRNA or control siRNA before being fixed, labeled with the NCC primary antibody followed by a 488 conjugated secondary antibody, and then imaged for fluorescence. The cells transfected with filamin A siRNA showed less phospho-NCC compared with the cells transfected with the control siRNA (Figure 5).

### 3.3. NCC and Filamin A Directly Interact in mDCT15 Cells

The actin cytoskeleton plays an important role in stabilizing transport proteins at the plasma membrane. Filamin A is an actin binding protein that anchors various transmembrane proteins and cytoplasmic proteins to the actin cytoskeleton. We examined whether filamin A directly interacts with the NCC. NCC protein was immunoprecipitated from mouse kidney cortex tissue lysates or mDCT15 cellular lysates before probing for a filamin A protein interaction using a specific filamin A antibody by a Western blot. The IP-Western showed an immunoreactive band above 250 kDa corresponding filamin A (Figure 6A). Similarly, filamin A protein was immunoprecipitated from mouse kidney cortex tissue lysates or mDCT15 cell lysates before probing for NCC protein interaction using a specific NCC antibody by Western blot. The IP-Western showed an immunoreactive band at approximately 130 kDa corresponding with the NCC (Figure 6A). In order to further confirm the direct interaction between the NCC and filamin A in the kidney, we subjected the eluent of an NCC immunoprecipitation experiment using mouse kidney cortex protein lysate to an LC-MS/MS analysis for protein identification. Multiple signature peptides corresponding with filamin A were detected in the NCC immunoprecipitation sample, as shown in Figure 6B.

### 3.4. CaMKII Inhibition Augments the Total NCC Protein Expression in mDCT15 Cells

Proteins associated with the actin cytoskeleton serve as an organizing center for transmembrane proteins and various other proteins including kinases, adaptor proteins, and cytoplasmic proteins. These actin cytoskeleton-associated proteins such as filamin A are thought to stabilize the membrane proteins, and the phosphorylation of filamin A results in a reorganization of the actin cytoskeleton. We investigated the NCC protein expression after inhibiting CaMKII, a kinase previously shown to phosphorylate filamin A. The pharmacological inhibition of filamin A with KN93 in mDCT15 cells resulted in an increase in the NCC protein expression in the membrane fractions (Figure 7).

### 3.5. CaMKII Inhibition Increases Phospho-NCC and Its Activity in mDCT15 Cells

Phosphorylation of the NCC is essential for its activity at the luminal plasma membrane of renal distal convoluted tubule cells. We investigated the density of phospho-NCC in the membrane fractions of mDCT15 cells treated with vehicle or the CaMKII inhibitor KN93. A Western blot and densitometric analysis showed that the phospho-NCC in the membrane fractions of mDCT15 cells was greater in cells treated with KN93 compared to vehicle (Figure 8A,B). In order to determine whether NCC activity was also affected by the pharmacological inhibition of CaMKII, we treated mDCT15 cells with either vehicle control or KN93 before performing Western blotting or thiazide-sensitive uptake assays. As shown in Figure 8C, NCC activity increased in a dose-dependent manner in mDCT15 cells treated with 0.05 μM, 0.5 μM, or 5 μM KN93 compared wit vehicle-treated cells. The density of filamin A and phospho-filamin A protein expression in the cytoplasmic and membrane fractions of mDCT15 cells was also assessed after KN93 treatment compared to MOCK (vehicle) treatment (Figure 9). 

## 4. Discussion

The cytoskeleton of eukaryotic cells provides structural support for the plasma membrane and contributes to many dynamic processes. Furthermore, it regulates numerous epithelial transport proteins including renal epithelial sodium channels (ENaC), aquaporin 2 (AQP2), and chloride channels [9,12,20,21,22,23]. Filamin A is an actin cytoskeleton protein that is able to regulate cell shape and motility [16,24]. It can crosslink actin and act as an intracellular signaling scaffold through binding to a variety of signaling network components [25].

Filamin A has been shown to interact with various transporters and ion channels and alter their function. Kim et al. showed that filamin A stimulates the surface expression of large-conductance Ca2^+^-activated K^+^ channels [26]. Sampson et al. showed a direct interaction between filamin A and the inwardly rectifying potassium channel, Kir2.1 [17]. TheLin et al. showed that filamin associates with the amino terminus of the cystic fibrosis transmembrane conductance regulator (CFTR) and stabilizes its expression at the plasma membrane [27]. We previously showed that the phosphorylation of filamin by CaMKII disrupts the cytoskeleton organization and attenuates the association between ENaC and MARCKS, leading to a decrease in channel activity [9].

The regulation of the renal NCC by the actin cytoskeleton has not been thoroughly investigated. One report by Dimke showed that the actin cytoskeletal protein γ-adducin directly binds to the amino terminus of the renal NCC in an unphosphorylated state and stimulates its activity [15]. Here, we identified a novel and functional association between the NCC and the actin cytoskeleton protein filamin A in mouse distal convoluted tubule cells (Figure 6).

We directly measured NCC activity after treating cells with two different disruptors of the actin cytoskeleton, latrunculin B and cytochalasin D. Latrunculin B inhibits the assembly of actin microfilaments [28,29] and cytochalasin D inhibits the association and dissociation of actin subunits after binding to the barbed ends of actin filaments [30]. Moreover, latrunculins have been shown to be more effective in the disruption of actin filaments and at lower concentrations compared with cytochalasin [31]. Both latrunculin B and cytochalasin D treatment caused a decrease in NCC activity compared with the control (Figure 1). The central hypothesis that we addressed in this study was that filamin A stabilizes NCC protein expression and activity at the apical plasma membrane in a mechanism dependent on CaMKII activity.

The phosphorylation of various amino acid residues within the amino-terminal domain of the NCC is associated with its activation. The phosphorylation of the NCC at Thr-53, Thr-58, and Ser-71 serves as a positive regulator of the NCC transport activity [32]. Our biochemical data were consistent with others that showed that the phosphorylated form of the NCC is associated with the plasma membrane. We corroborated these studies by performing thiazide-sensitive ^22^Na^+^ uptake assays in order to measure the NCC function after the inhibition of the CaMKII activity in mDCT15 cells. The physiological significance of the interaction between filamin A and the NCC in the aldosterone-sensitive distal nephron can be appreciated at two levels. First, the function of the NCC at the luminal membrane of cells in the late part of the distal convoluted tubule (DCT2) is dependent on the NCC being phosphorylated at the intracellular amino terminus. Assuming filamin A binds the NCC in a similar manner as γ-adducin, this interaction could further increase the phosphorylation and, hence, the membrane expression of the NCC. It is likely that filamin and γ-adducin do not compete for the same binding site but instead form a complex that serves as an organizing center for regulatory protein kinases to be in close proximity to the phosphorylation sites of the NCC for its phosphorylation and activation. For example, the serine/threonine WNK family of kinases activates the intermediate kinases ste20-related proline/alanine-rich kinase (SPAK) and oxidative stress-responsive kinase 1 (OSR1), which in turn phosphorylate the NCC [33]. The prediction algorithm NetPhos3.1 revealed multiple CaMKII phosphorylation sites for a human NCC. This suggests that CaMKII not only indirectly regulates the NCC through filamin A but it may also directly phosphorylate and regulate NCC activity.

The phosphorylation of filamin may function as a molecular switch for NCC activity and perhaps serve as a physiologically important feedback mechanism (Figure 10). The binding between filamin A and the NCC may attenuate the interaction between the NCC and NEDD4-2, thus reducing NCC degradation. The association between filamin and the NCC may also potentiate protein–protein interactions with other signaling proteins to allow for a further regulation of sodium and chloride transport in the DCT2.

The focus of this study was to determine the role of CaMKII in the regulation of filamin A and the NCC membrane expression and activity in mDCT15 cells. However, filamin A is also a substrate of the serine/threonine kinase p21-activated kinase 1 (Pak1) [34], the ribosomal S6 kinase (RSK) [35], cyclin B1/Cdk1 [36], and akt [37]. Future studies should be performed to determine whether or not these kinases also regulate the NCC membrane expression and activity in a filamin-dependent manner.

These results corroborated the role of the actin cytoskeleton and associated proteins in regulating the NCC function in the distal convoluted tubule. As the renal NCC plays an important role in maintaining the total body salt and blood pressure homeostasis, these data suggest that CaMKII and filamin A may play an indirect role in regulating blood pressure by modulating the renal NCC. This warrants additional studies to investigate the regulation of the NCC by CaMKII and filamin A in the native kidney. Furthermore, the NCC is expressed in other tissues including the intestine and additional studies may be performed to investigate its regulation by filamin and CaMKII.

## 5. Conclusions

Taken together, our data identified a novel and functional association between the actin cytoskeleton protein filamin A and the renal NCC in cultured mouse distal convoluted tubule cells and in the native kidney cortex. The pharmacological inhibition of CaMKII resulted in an increase of the functional form of the NCC (phospho-NCC) at the luminal membrane. Conversely, CaMKII phosphorylates filamin A causing a reorganization of the actin cytoskeleton and a decrease in the NCC membrane expression and activity as well as its association with the actin cytoskeleton. Future studies are necessary to determine whether the association between the NCC and filamin A are augmented in hypertension and if this association inhibits the recycling or degradation of the NCC through the proteasomal pathway.

## Figures and Tables

**Figure 1 biology-10-01335-f001:**
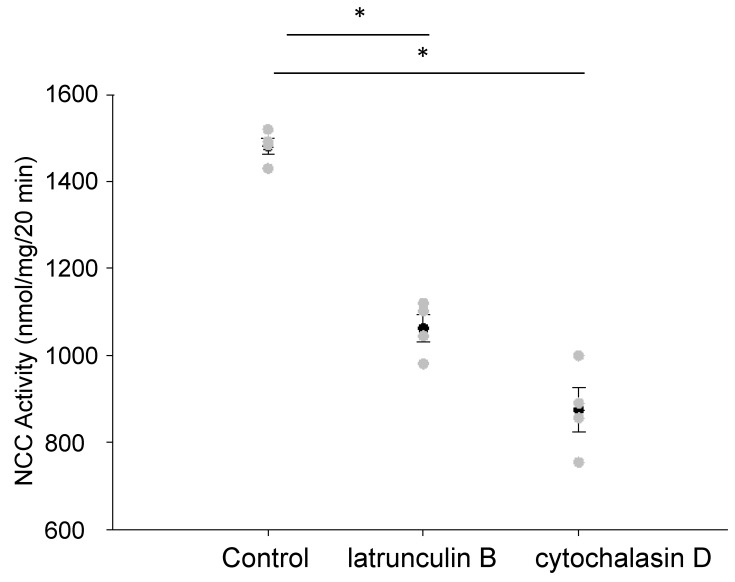
Effect of disrupting the actin cytoskeleton on NCC activity. The mDCT15 cells treated with latrunculin B (500 nM) or cytochalasin D (1 μM) for 30 min resulted in a decrease in NCC activity compared with the cells treated with the vehicle alone (control). Thiazide-sensitive ^22^Na^+^ uptake was calculated as the difference between the uptakes with and without thiazide. Four independent experiments were performed, N = 4. * represents p < 0.05.

**Figure 2 biology-10-01335-f002:**
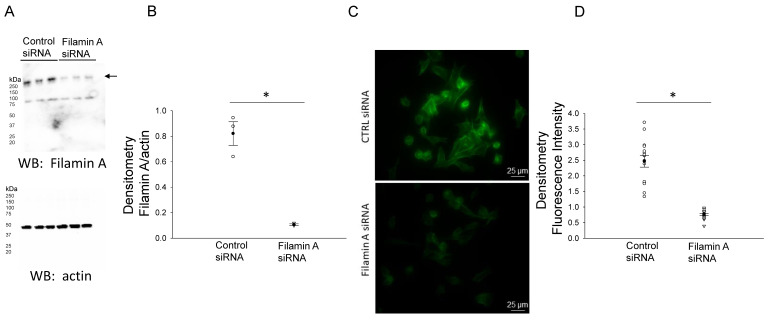
Western blot and densitometric analysis of Filamin A after siRNA mediated knockdown of filamin A. (**A**) Western blot of Filamin A and actin after mDCT15 cells were transiently transfected with filamin A siRNA or control siRNA and then harvested for protein. (**B**) Densitometric analysis of the immunoreactive filamin A band normalized to actin from panel A. (**C**) Immunocytochemistry image of mDCT15 cells probed for filamin A flurescence using a rabbit polyclonal filamin A primary antibody and an antirabbit 488 conjugated secondry antibody. (**D**) Densitometric analysis of the images in panel (**C**). Three independent experiments were performed, N = 3. * represents p < 0.05.

**Figure 3 biology-10-01335-f003:**
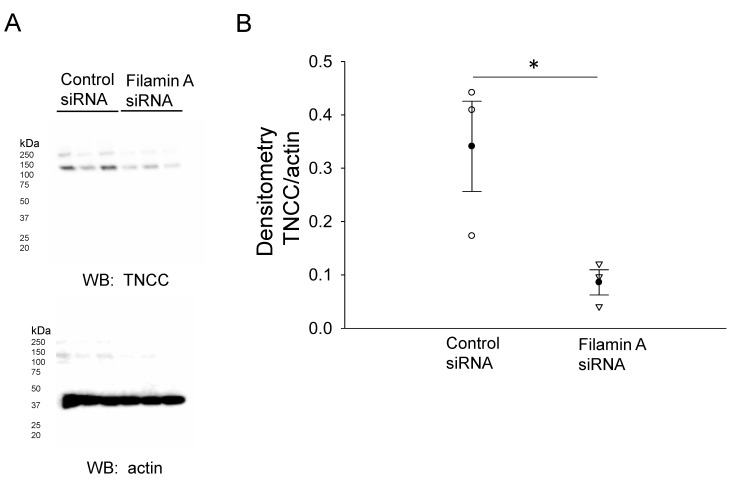
Western blot analysis of the total NCC in mDCT15 cells transfected with filamin A siRNA or control siRNA. (**A**) Western blot of the total NCC (TNCC) protein in mDCT15 cells after the siRNA-mediated knockdown of filamin A compared with the control. (**B**) Densitometric analysis of the immunoreactive bands in (**A**). Three independent experiments were performed, N = 3. * represents *p* < 0.05.

**Figure 4 biology-10-01335-f004:**
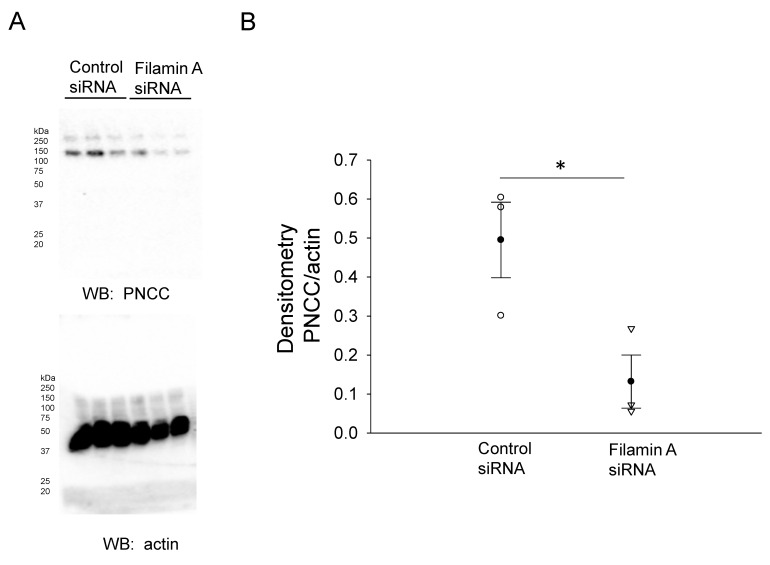
Western blot and densitometric analysis of pNCC after siRNA mediated knockdown of filamin A. (**A**)Western blot of phospho-NCC (**top**) and actin (**bottom**) after mDCT15 cells were transiently transfected with filamin A siRNA or control siRNA and then harvested for protein. (**B**) Densitometric analysis of the immunoreactive Phospho-NCC band normalized to actin from panel A. Three independent experiments were performed, N = 3. * represents p < 0.05.

**Figure 5 biology-10-01335-f005:**
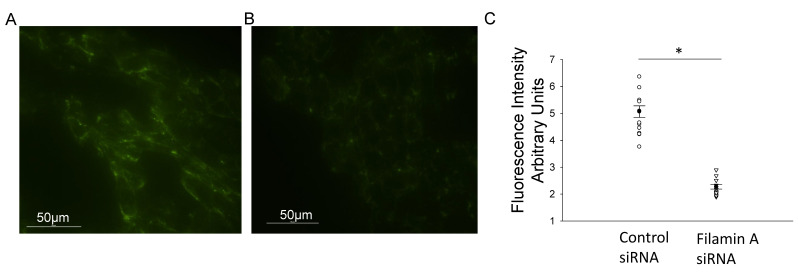
Immunocytochemistry of mDCT15 cells transfected with filamin A siRNA or control siRNA and then probed for phospho-NCC immunocytochemistry for mDCT15 cells transfected with control siRNA (**A**) or Filamin A siRNA (**B**). (**C**) Densitometric analysis of the positive staining in panels (**A**,**B**). Three independent experiments were performed, N = 3. * represents *p* < 0.05.

**Figure 6 biology-10-01335-f006:**
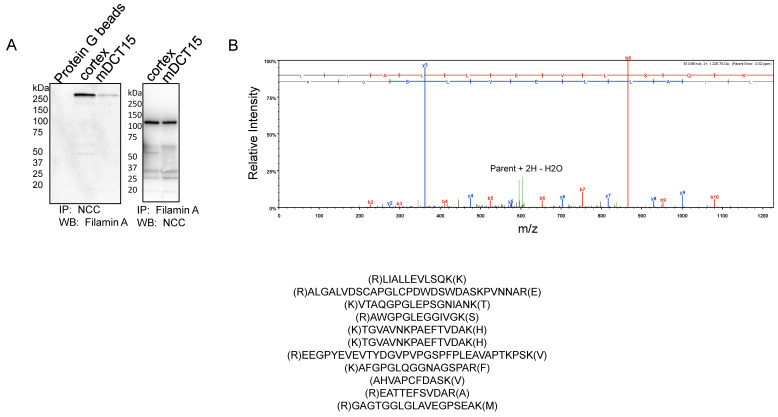
Association between filamin A and the NCC in the kidney cortex and in mDCT15 cells. (**A**) Immunoprecipitation and Western blot analysis probing for a direct interaction between filamin A and the NCC. A total NCC antibody was used to immunoprecipitate endogenous NCC and putative binding proteins (left panel) and the filamin A antibody was used to immunoprecipitate endogenous filamin A protein and putative binding proteins (right panel) from the kidney cortex tissue lysates and mDCT15 cell lysates. The eluents were subjected to SDS-polyacrylamide gel electrophoresis before being probed by a Western blot for filamin A protein (left panel) or the NCC (right panel) using specific antibodies. IP refers to immunoprecipitation; WB refers to Western blotting. (**B**) LC-MS/MS analysis identifying filamin as a novel protein binding partner for the NCC. The eluents from the NCC immunoprecipitated complexes were resolved by SDS-PAGE and the gels were Coomassie stained. The gel band corresponding with the immunoreactive filamin band observed in the Western blot in panel (**A**) was excised and subjected to trypsin digestion and a subsequent LC-MS/MS analysis. The top panel is a representative spectrum with unique filamin A peptides identified by LC-MS/MS. The signature peptides corresponding with filamin A are listed in the bottom panel.

**Figure 7 biology-10-01335-f007:**
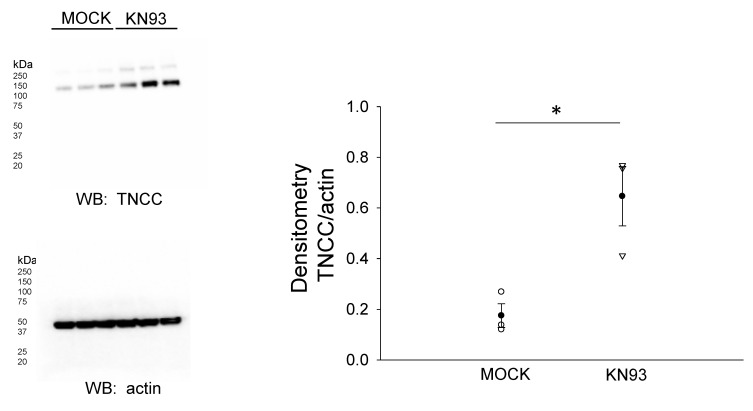
Effect of the CaMKII inhibitor KN93 on total NCC (TNCC) protein expression. (**A**) Western blot from the samples of three independent experiments (N = 3) after mDCT15 cells were treated with the vehicle (MOCK) or the CaMKII inhibitor KN93 for 4 h before harvesting the cells for protein and isolating the membrane fraction. (**B**) Summary bar graph of the densitometric analysis for the Western blot in panel (**A**). The band intensities were quantified after normalizing the immunoreactive NCC bands to actin. * represents *p* < 0.05.

**Figure 8 biology-10-01335-f008:**
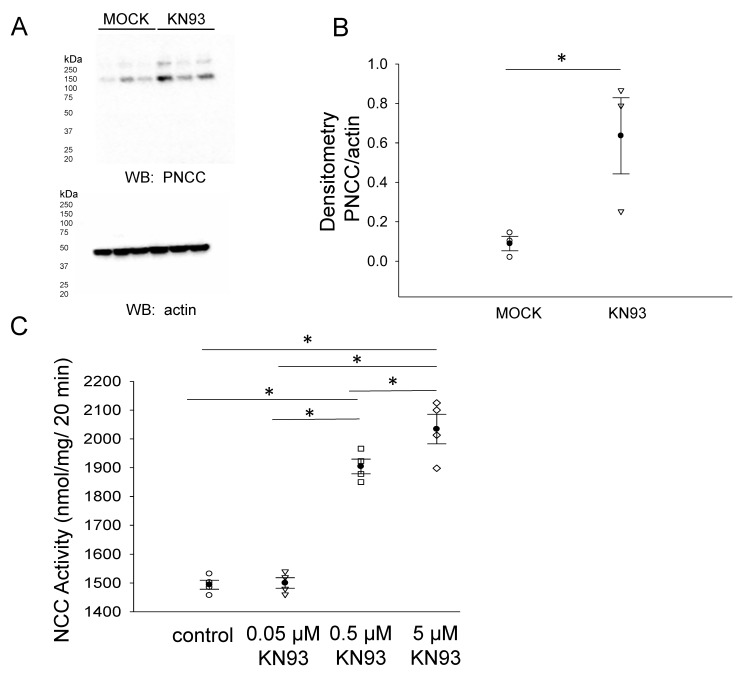
Effect of CaMKII inhibition on phospho-NCC protein expression and NCC activity. (**A**) Western blot from the samples of three independent experiments (N = 3) in which mDCT15 cells were treated with the vehicle (MOCK) or the CaMKII inhibitor KN93 for 4 h before harvesting the cells for protein and isolating the membrane fraction. Blots probed with a phospho-NCC-specific antibody showed an increase in the phospho-NCC protein expression. (**B**) Summary bar graph of the densitometric analysis for the Western blot in panel (**A**). The band intensities were quantified after normalizing the immunoreactive bands in the Western blot for phospho-NCC to a Western blot for actin. (**C**) The effect of the pharmacological inhibition of CaMKII, KN93, on NCC activity in mDCT15 cells. The cells were treated with either the vehicle control (DMSO), 0.05 μM KN93, 0.5 μM KN93, or 5 μM KN93 for 4 h prior to the measurement of the thiazide-sensitive ^22^Na^+^ uptake. The thiazide-sensitive ^22^Na^+^ uptake was calculated as the difference between the uptakes with and without thiazide. Four independent experiments were performed, N = 4. * represents *p* < 0.05.

**Figure 9 biology-10-01335-f009:**
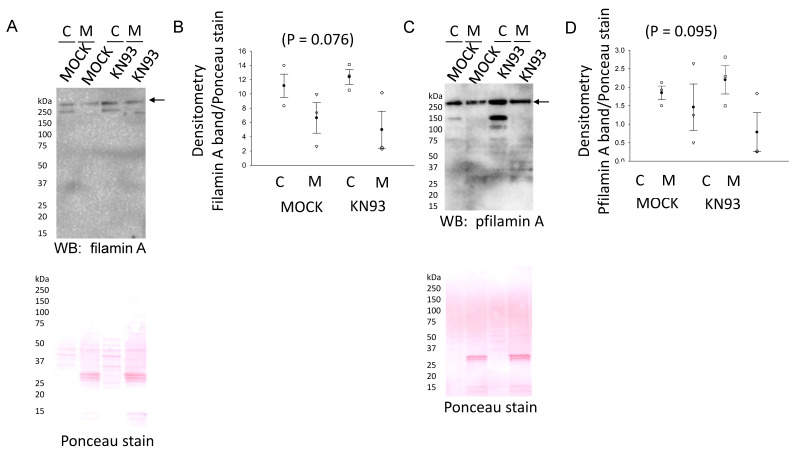
Filamin A protein expression and phosphorylation after KN93 treatment in mDCT15 cells. (**A**) Western blot analysis of filamin A protein expression in the cytoplasmic (**C**) and membrane (M) fractions of mDCT15 cells. (**B**) Densitometric analysis of the filamin A immunoreactive bands indicated by an arrow in (**A**). (**C**) Western blot analysis of the phospho-filamin A (pfilamin A) protein expression in the cytoplasmic (**C**) and membrane (M) fractions of mDCT15 cells. (**D**) Densitometric analysis of the filamin A immunoreactive bands indicated by an arrow in (**C**).

**Figure 10 biology-10-01335-f010:**
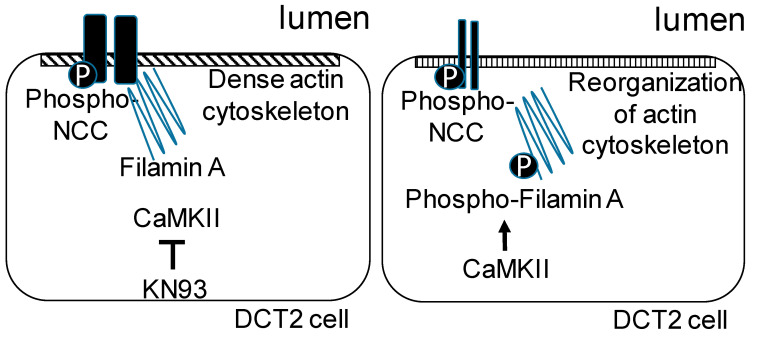
Proposed model for the regulation of the NCC by filamin A and CaMKII. Phosphorylation of filamin A by CaMKII results in a reorganization of the actin cytoskeleton and low basal NCC activity at the luminal membrane of a DCT2 cell (right cell). A deficiency in CaMKII or the pharmacological inhibition of CaMKII by KN93 blocks the phosphorylation of filamin A, resulting in a dense actin cytoskeleton and an accumulation of the active phosphorylated form of the NCC at the luminal membrane (left cell).

## Data Availability

The data from this study are presented within the figures.

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
