# Peer review of "The Pharmacological Inhibition of CaMKII Regulates Sodium Chloride Cotransporter Activity in mDCT15 Cells"

_biology, 2021, doi:10.3390/biology10121335_

Round 1

Reviewer 1 Report

The main findings of the present study are: 1) disruption of cytoskeleton decreased thiazide-sensitive Na uptake and NCC/pNCC expression in mDCT15 cell line ; 2) immunoprecipitation of DCT cell lysates with NCC antibody pulls down actin cytoskeleton protein Filamin A ; 3) pharmacological inhibition of CaMKII increase the expression of NCC and pNCC and thiazide-sensitive Na uptake. The finding that cytoskeleton and NCC interaction affects NCC activity is novel, but it is not surprised because many membrane transporter proteins are known to interact with cytoskeleton. However, the interpretation that CaMKII effect on NCC is due to CaMKII-mediating Filamin A phosphorylation is not supported by any data. Authors need more work to support this hypothesis. Also, all key experiments were performed in mDCT15 cell lines. This reviewer realizes that some experiments would be difficult to preform in animals. However, author could easily examine the expression pNCC expression in the mouse DCT using immunofluorescence microscope in the kidney treated with cytochalasin D. This would enhance the auditor’s conclusion.   

Specific Points:

1.Since WNK is a major regulator for NCC, what is the role of WNK in mediating Filamin A-induced regulation of NCC activity?

  1. Is it possible whether CaMKII directly affect the activity of NCC without filamin A involvement?

3 Any explanation why inhibition of Filamin A using siRNA caused a larger decrease NCC  ( Figure 2) than pNCC (Figure 3). NCC degradation?

4 In figure 8, author only provides DCT2 cell model. Dose the regulation of NCC by filamin A and CaMKII only take place in the DCT2?

Author Response

We thank this reviewer for the helpful comments and suggestions. We have addressed each point as outlined below.

Reviewer 1:

The main findings of the present study are: 1) disruption of cytoskeleton decreased thiazide-sensitive Na uptake and NCC/pNCC expression in mDCT15 cell line ; 2) immunoprecipitation of DCT cell lysates with NCC antibody pulls down actin cytoskeleton protein Filamin A ; 3) pharmacological inhibition of CaMKII increase the expression of NCC and pNCC and thiazide-sensitive Na uptake. The finding that cytoskeleton and NCC interaction affects NCC activity is novel, but it is not surprised because many membrane transporter proteins are known to interact with cytoskeleton. However, the interpretation that CaMKII effect on NCC is due to CaMKII-mediating Filamin A phosphorylation is not supported by any data. Authors need more work to support this hypothesis. Also, all key experiments were performed in mDCT15 cell lines. This reviewer realizes that some experiments would be difficult to preform in animals. However, author could easily examine the expression pNCC expression in the mouse DCT using immunofluorescence microscope in the kidney treated with cytochalasin D. This would enhance the auditor’s conclusion.   

Specific Points:

1.Since WNK is a major regulator for NCC, what is the role of WNK in mediating Filamin A-induced regulation of NCC activity?

In this manuscript, we did not investigate whether the WNK-OSR1 and SPAK signaling cascade is regulated by Filamin A and CaMKII or whether it regulates filamin-CaMKII mediated regulation of NCC activity.  To our knowledge the role of WNK in mediating Filamin A induced regulation of NCC is unknown and has not been studied.  We plan to investigate his pathway along with the Nedd4-2 pathway in a future study.

  1. Is it possible whether CaMKII directly affect the activity of NCC without filamin A involvement?

The prediction algorithm NetPhos3.1 did reveal multiple CaMKII phosphorylation sites for the human NCC protein sequence.   These sites include:  AELPTTETP, ELPTTETPG, PTTETPGDA, PGDATLCSG, ALCSGRFT, SGRFTISTL, RFTISTLLS, FTISTLLSS, STLLSSDEP, TLLSSDEPS, SDEPSPPAA, AAYDSSHPS, AYDSSHPSH, SSHPSHLTH, PSHLTHSST, HLTHSSTFC, LTHSSTFCM, THSSTFCMR, and others.  This suggest CaMKII not only indirectly regulates NCC through the regulation of filamin A, but it may directly phosphorylate and regulate NCC activity.  We have included a sentence discussing this analysis in the discussion section.  In order to thoroughly investigate the direct role of CaMKII in the regulation of NCC activity, reporter constructs will need to be created and a series of experiments including site directed mutagenesis and phospho-enrichment proteomic studies can be performed to further investigate and identify the amino acid residues involved.   The authors plan to address this topic in the future.  

3 Any explanation why inhibition of Filamin A using siRNA caused a larger decrease NCC (Figure 2) than pNCC (Figure 3). NCC degradation?

We repeated both experiments for TNCC (new Figure 3) and pNCC (new Figure 4) and observed similar decreases in the amount of total-NCC (from 0.34+/-0.08 in control siRNA transfected cells compared to 0.09+/-0.02 in filamin A siRNA transfected cells) and phospho-NCC (from 0.50+/-0.10 in control siRNA transfected cells compared to 0.13+/-0.07 in filamin A siRNA transfected cells).

4) In figure 8, author only provides DCT2 cell model. Dose the regulation of NCC by filamin A and CaMKII only take place in the DCT2?

We are aware of NCC expression in other tissues including the intestine (PMID: 15781471. https://pubmed.ncbi.nlm.nih.gov/15781471/; PMID: 17656470. https://pubmed.ncbi.nlm.nih.gov/17656470/)  But, the focus of our study was the regulation of NCC by filamin A and CaMKII in the DCT2.  We have added a sentence to the discussion section to mention that NCC expression is not exclusive to the DCT2 and additional studies are necessary to determine its regulation by this mechanism in other tissues.

Reviewer 2 Report

The aim of this manuscript is to investigate the interaction between Filamin A and renal NCC. They also studied the potential role of Filamin A in regulation of NCC in kidney membrane. The author showed that destroyed actin cytoskeleton by latrunculin B and cytochalasin D resulted in decrease NCC activity in mouse distal convoluted tubule cell line (mDCT15). Knockdown of Filamin A in the same cell line by siRNA not only suppress NCC protein expression but also inhibit NCC phosphorylation. Additionally, the author demonstrated potential direction interaction between Filamin A and NCC by using Immunoprecipitation method. Finally, the authors showed that phosphorylation of Filamin A by CAMKII resulted in decrease NCC expression, protein phosphorylation and activity in vitro.

This study is interesting as they present newly identified functional association between filamin A and renal NCC. They also showed a potential mechanism for NCC regulation and a potential drug candidate for hypertension.

Minor comments:

  1. in Fig. 1, is there any marker to quantify the degree of actin cytoskeleton destruction after latrunculin B or cytochalasin D treatment?
  2. In Fig. 2 and Fig. 3, Western blot analysis of filamin A expression in control and siRNA treated cells should be included as a good control.
  3. Similar as comments #2, immunostaining of filamin A in in control and siRNA treated cells should be presented.
  4. In Fig. 6 or 7, filamin A expression and phosphorylation in control and KN93 treated cells should be presented. 

Author Response

We thank this reviewer for the helpful comments and suggestions. We have addressed each point as outlined below.

Reviewer 2:

The aim of this manuscript is to investigate the interaction between Filamin A and renal NCC. They also studied the potential role of Filamin A in regulation of NCC in kidney membrane. The author showed that destroyed actin cytoskeleton by latrunculin B and cytochalasin D resulted in decrease NCC activity in mouse distal convoluted tubule cell line (mDCT15). Knockdown of Filamin A in the same cell line by siRNA not only suppress NCC protein expression but also inhibit NCC phosphorylation. Additionally, the author demonstrated potential direction interaction between Filamin A and NCC by using Immunoprecipitation method. Finally, the authors showed that phosphorylation of Filamin A by CAMKII resulted in decrease NCC expression, protein phosphorylation and activity in vitro.

This study is interesting as they present newly identified functional association between filamin A and renal NCC. They also showed a potential mechanism for NCC regulation and a potential drug candidate for hypertension.

Minor comments:

1) In Fig. 1, is there any marker to quantify the degree of actin cytoskeleton destruction after latrunculin B or cytochalasin D treatment?

In a published study from our group we performed immunocytochemistry studies to investigate the degree of actin cytoskeleton destruction by cytochalasin treatment.   (Figure 3:  Am J Physiol Renal Physiol. 2014 Jul 1;307(1):F86-95. PMID: 24829507.  Specific markers may include the organization of actin associated proteins including spectrin, MARCKS, filamin, fodrin, and the actin cytoskeleton linker protein ezrin.

2) In Fig. 2 and Fig. 3, Western blot analysis of filamin A expression in control and siRNA treated cells should be included as a good control.

We performed the suggested experiment and have now included a Western blot and the densitometric analysis (new Figure 2A) showing less filamin A protein expression in cultured mDCT15 cells after siRNA transfection of filamin A compared to control siRNA.

3) Similar as comments #2, immunostaining of filamin A in in control and siRNA treated cells should be presented.

We performed the suggested experiment and have now included the immunostaining of filamin A in control and siRNA transfected cells (new Figure 2B).

4) In Fig. 6 or 7, filamin A expression and phosphorylation in control and KN93 treated cells should be presented. 

As suggested, we have included additional data (Figure 10) showing filamin A expression and phosphorylation for the control and KN93 groups. 

Round 2

Reviewer 1 Report

No more comments